

# Data-Driven Scaling Methods for Soil Moisture Cosmic Ray Neutron Sensors

Roland Baatz[1], Patrick Davies[2], Paolo Nasta[3], Heye Bogena[4]

[1] Leibniz Centre for Agricultural Landscape Research (ZALF), Eberswalder Str. 84, 15374 Muncheberg, Germany.
[2] Department of Meteorology and Climate Science, KNUST, Kumasi-Ghana.
[3] Department of Agricultural Sciences, Division of Agricultural, Forest and Biosystems Engineering, University of Naples Federico II, Portici (Naples), Italy.
[4] Institute of Bio- and Geosciences, Agrosphere (IBG-3), Forschungszentrum Jülich GmbH, 52428 Jülich, Germany.

*Correspondence to*: Roland Baatz (roland.baatz@zalf.de)

**Abstract**

Cosmic ray neutron probes (CRNS) are increasingly used for soil moisture measurement, yet uncertainties persist due to reliance on traditional analytical scaling methods that may not fully account for site-specific and sensor-specific characteristics. This study introduces a novel, data-driven calibration approach to estimate key scaling parameters (beta, psi, and omega) for CRNS, emphasizing local environmental factors and sensor attributes. The method provides a more flexible, empirical
approach to calibration by directly calculating correction parameters from measurement data.

The results demonstrate that the new method is both reliable and robust, showing strong correlations between the estimated parameters and those predicted by analytical methods. However, the study also reveals systematically higher variability in calibration parameters than previously assumed, underscoring the importance of data quality and careful selection of NMDB
reference sites. Sensor-specific factors, such as the energy spectrum, along with site-specific factors like elevation and geographic proximity to NMDB sites, significantly influence scaling parameters, highlighting the necessity for site- and sensor-specific calibration to improve soil moisture estimates. Future research should focus on refining these scaling methods and enhancing data quality to further improve CRNS measurement accuracy.



## 1    Introduction


Soil moisture describes the quantity of water present in the vadose zone. Soil moisture or soil water content has significant impacts on a number of soil properties, including thermal and hydraulic soil characteristics, groundwater recharge processes, infiltration rates, the availability of water for plants, irrigation requirements, and the severity of drought conditions. (Řehoř et al., 2024; Humphrey et al., 2021; Vereecken et al., 2008). In order to effectively manage these critical processes and make

informed decisions, soil moisture measurements have been developed at various scales, ranging from the pore scale to the plot scale, field scale, and global scale (Robinson et al., 2008). Pore and plot scale measurements primarily utilize the geoelectrical properties soils (Dorigo et al., 2021; Robinson et al., 2008), while field scale measurements often rely on networks of point scale sensors (Korres et al., 2015; Dorigo et al., 2021) or nuclear physics principles (Zreda et al., 2008). From the regional to the global scale, soil moisture is usually quantified by analyzing the dielectric properties of the soil using passive or active

microwave sensors (Manfreda et al., 2018; Entekhabi et al., 2010).

Cosmic Ray Neutron Sensors (CRNS) provide a critical link between small-scale and large-scale soil moisture measurements, bridging the gap from local field measurements to broader regional assessments (Zreda et al., 2008; Baatz et al., 2014). CRNS operate by detecting epithermal neutrons generated by cosmic rays interacting with the Earth's atmosphere. The hydrogen in

soil water plays an important role in attenuating epithermal neutrons in the lower atmosphere. By measuring the epithermal neutrons above the soil with the CRNS, it is therefore possible to estimate the average soil moisture over an area of several tens of hectares. (Köhli et al., 2015; Desilets et al., 2010). This unique capability allows CRNS to integrate spatial variability in soil moisture across a landscape more effectively than point scale soil moisture sensors. Furthermore, it serves to complement satellite-based measurements, which cover larger scales but with lower resolution (Babaeian et al., 2019; Montzka

et al., 2017). As such, improving soil moisture estimates from CRNS offers significant potential for enhancing water resource management, agricultural practices, and drought monitoring by providing reliable, intermediate-scale data (Baatz et al., 2017; Brogi et al., 2022).

Appropriate signal processing of CRNS raw data is crucial for the accurate conversion of neutron count rates to soil moisture

(Davies et al., 2022). In addition to hydrogen within the CRNS footprint, the CRNS neutron signal is influenced by various other factors, including atmospheric pressure, air humidity, and incoming neutron intensity. These factors are typically accounted for by applying scaling functions for atmospheric pressure, air humidity and incoming neutron intensity to isolate the neutron signal stemming from hydrogen (Desilets and Zreda, 2003; Desilets et al., 2006). For instance, scaling for atmospheric pressure is necessary because higher pressure compresses the atmosphere, increasing neutron attenuation and

reducing detected counts (Zreda et al., 2012; Baatz et al., 2014). This requires applying a correction factor to normalize neutron flux to a standard pressure. Similarly, air humidity affects the number of epithermal neutrons detected by increasing the amount of hydrogen in the air, necessitating a separate humidity correction to ensure accurate soil moisture estimation (Rosolem et al.,



2013; Köhli et al., 2021). In addition, incoming neutron intensity varies over time due to solar activity and cosmic events, requiring adjustments to the neutron count rates to account for these fluctuations (Gerontidou et al., 2021; Hawdon et al., 2014; McJannet and Desilets, 2023). Additionally, model-data fusion techniques that integrate CRNS signals with other measurements and model predictions of soil moisture are increasingly used to refine soil moisture estimates (Baatz et al., 2017; Li et al., 2024). The improvement of these signal processing methods is of paramount importance for the enhancement of the accuracy, reliability, and resolution of soil moisture data obtained from CRNS (Davies et al., 2022; Brogi et al., 2022). Ultimately, this will facilitate more informed decisions in agricultural and hydrological management, as well as more accurate observations of climate change effects (Bogena et al., 2022).

While CRNS are increasingly used for soil moisture measurement (Bogena et al., 2022; Zreda et al., 2012), significant uncertainties persist due to the reliance on traditional analytical approaches for correcting environmental factors such as atmospheric pressure, humidity, and incoming neutron intensity. For example, scaling for air humidity was found to effect CRNS neutron intensities linearly (Rosolem et al., 2013) or even steeper (Köhli et al., 2021). Relation of incoming neutron intensity depends on CRNS sensor and reference site of the monitor observing incoming neutron intensity (McJannet and Desilets, 2023; Hawdon et al., 2014). For atmospheric pressure, scaling coefficients are site specific and depend on cutoff rigidity and elevation of the CRNS (Tirado-Bueno et al., 2021; Desilets et al., 2006). All methods often depend on generalized scaling functions based on global estimates, such as cutoff rigidity and other global relationships, which may not accurately reflect local site characteristics, sensor manufacturing attributes, or sensor-specific energy spectra that can influence calibration parameters. Often, these methods have been developed based on available data from incoming neutron or cosmic ray monitors of the NMDB project for the reason of high data availability from these monitors and high signal-to-noise ratio with CRNS observations. However, the difference in sensor characteristics and objective to detect soil moisture with CRNS led to the critical need to develop more site-specific and sensor-specific scaling approaches that account for the often very sensor specific and local conditions.

This study presents a novel data-driven, empirical approach that allows the analysis of uncertainties in soil moisture estimation without relying on the assumptions embedded in traditional Monte Carlo neutron transport codes. By considering soil moisture dynamics as noise term and directly calculating correction parameters from the measurement signal, this inverse modelling method offers a new perspective on calibration. The approach allows for a detailed evaluation of the accuracy of current scaling functions, the selection of NMDB monitors, and the impact of local environmental factors on calibration. By improving the methods for processing CRNS signals, this study aims to enhance the accuracy and reliability of soil moisture data, ultimately supporting better-informed decisions in agriculture, hydrology, and climate monitoring.





## 2      Methods

### 2.1      Theoretical aspects

#### 2.1.1      Scaling with atmospheric pressure

Neutron flux was found to be exponentially dependent on atmospheric pressure (Tirado-Bueno et al., 2021; Desilets et al., 2006; Desilets and Zreda, 2003):

$$N_{ref}/N_1 = exp\left(beta\left(P_1 - P_{ref}\right)\right) \tag{1}$$

where *beta* is a constant proportional to the attenuation length. Beta and the attenuation length scale the reference neutron flux $N_{ref}$ observed under reference atmospheric pressure $P_{ref}$ in hPa to observed neutron flux $N_1$ at time $t = 1$ given observed atmospheric pressure $P_1$ at time $t = 1$ in hPa. Noteworthy is, the scaling factor is exponential and consistent across different atmospheric pressures, meaning the neutron intensity scales equally for any pressure difference. This is different for the two following scaling approaches for air humidity and incoming neutron flux. A second noteworthy characteristic is that with a very small beta, such as −0.0076, the scaling becomes nearly linear. The physical explanation of the scaling relationships has been widely studied and discussed (Tirado-Bueno et al., 2021; Schrön et al., 2024; Nuntiyakul et al., 2014; Clem and Dorman, 2000). Most of these analyses focused on neutron monitors, with only a limited number of analyses using CRNS, which measure neutron flux at the energy spectrum relevant for soil moisture detection (Schrön et al., 2024). Here, we focus on epithermal neutron count data from twelve CRNS stations from the COSMOS-Europe data set (Bogena et al., 2022).

### 2.1.2      Scaling with incoming neutron intensity

The second dependency of neutron flux observed is that on incoming neutron intensity. Here, commonly a linear scaling approach is adopted to account for the relative change of incoming neutron intensity (Baatz et al., 2015; Zreda et al., 2012; Hawdon et al., 2014). Reference stations are those of the neutron monitor database nmdb.eu (Bütikofer, 2018; Gerontidou et al., 2021). The scaling depends on the location of the cosmic ray neutron sensor along the geomagnetic cutoff rigidity, longitude and latitude of the earth, elevation, and energy spectrum observed of either sensor amongst potentially other factors. Recently, new generalized relationships were established for CRNS by McJannet and Desilets (2023). Here, we adopt the linear scaling approach previously adopted because of its robustness:

$$N_{ref}/N_1 = \left(1 + psi \times \left(Inc_1 - Inc_{ref}\right)\right) \tag{2}$$

where psi is a constant specific to the cosmic ray neutron sensor, its location, manufacturing and measurement characteristics, and the neutron monitor used for incoming neutron intensity. $Inc_{ref}$ is the reference incoming neutron intensity, $Inc_1$ is the neutron intensity at the time of observation $t = 1$, and $N_{ref}$ is the reference neutron flux observed. Incoming neutron intensity is calculated as the ratio of incoming neutron count rate divided by the mean of the incoming neutron count rate over a time interval. Noteworthy is, when scaling based on incoming cosmic ray intensity, the scaling is linear, and the choice of reference intensity ($Inc_{ref}$) affects the result. Consequently, using different reference values leads to small inconsistencies in scaling,



causing the adjusted neutron intensities ($N_1$) to vary for different incoming neutron intensities. Although negligible for a small

range of $Inc_1 - Inc_{ref}$, it highlights the necessity of employing a mean of incoming neutron flux in lieu of an $Inc_{ref}$ at either end

of the $Inc$ spectrum over the measurement period. This is an important difference to Eq. 1, where scaling is consistent for

different reference values. Moreover, numerous studies have indicated that incoming neutron flux is depending on the cutoff

rigidity, which is why the position of the CRNS and NMDB stations should be as close as possible (Hawdon et al., 2014;

McJannet and Desilets, 2023). Here, we used six stations of the NMDB database with well comparable pair-wise cutoff

rigidities and a range of 0.65 to 8.53 Giga Volt (GV, Table 1).

**Table 1: NMDB sites used for correction of incoming neutron intensity with cutoff rigidity in Giga Volt (GV) calculated by (Gerontidou et al., 2021).**

| City/location | NMDB site | Country | Cutoff Rigidity [GV] | Altitude [m] |
|---|---|---|---|---|
| Apatity | Apty | Russia | 0.65 | 181 |
| Oulu | Oulu | Finland | 0.81 | 15 |
| Lomnicky Stit | Lmks | Slovakia | 3.84 | 2634 |
| Jungfraujoch | Jung1 | Switzerland | 4.49 | 3570 |
| Mexico | Mxco | Mexico | 8.28 | 2274 |
| Athens | Athn | Greece | 8.53 | 260 |


### 2.1.3   Scaling with air humidity

Rosolem et al. (2013) identified a linear relationship of air humidity and epithermal neutron via Monte Carlo neutron particle

simulations using the MCNPx model:

$$N_{ref}/N_1 = \left(1 + omega \times \left(abs_1 - abs_{ref}\right)\right) \tag{3}$$

where $abs_{ref}$ is the absolute reference absolute air humidity, i.e., water content in g m$^{-3}$ at two meters above ground, omega is

a constant and $abs_1$ is the air humidity in g m$^{-3}$ at the time of observation of neutron flux $N_h$. Again, the rate of change in $N_{ref}/N_1$

is not independent of $abs_{ref}$ chosen and leads to small discrepancies for different $abs_{ref}$. This is an important constraint and

strong reason to choose $abs_{ref}$ as mean over the measurement interval. While this scaling approach was confirmed in some

studies (Schrön et al., 2024), other studies have also indicated that air humidity could have a larger impact on neutron intensity

(Köhli et al., 2021).

### 2.1.4   Temporal aggregation of neutron flux

Scaling parameters beta, omega and psi were identified to be constant in time for a specific site except for little variation due

to changes in the solar spectrum (McJannet and Desilets, 2023; Dunai, 2000; Desilets and Zreda, 2003). The neutron flux data





follows a Poisson distribution as it is counts per time interval. For aggregating temporal Poisson data, it is advisable to use the
mean instead of the median over a specific time interval because the relationship between mean and cumulative sum over a
large time interval is proportional. Importantly, the standard deviation in relative terms decreases with increasing measurement
period because it is proportional to the square root of the number of counts. Therefore, aggregation over a prolonged time
interval is advantageous for reducing measurement uncertainty, although this approach inevitably entails a compromise in that
changes in other environmental variables over the measurement period cannot be directly accounted for.

## 2.2   Inversion of scaling functions

In this study, we employ an inverse estimation methodology to derive beta, omega, and psi values for each site within the
tested data set. This approach differs from previous studies that have utilized analytical techniques to ascertain the scaling
parameters. Our analysis draws upon atmospheric pressure, air humidity and epithermal neutron data of the European
COSMOS network (Bogena et al., 2022). Moreover, data from six neutron monitors were utilized from the NMDB database
(Table 1). All data were provided in hourly resolution, and quality checks were implemented to ensure the integrity of the data
set. Only data that fell within the physical range of the observed quantity were selected, and values that differed from
neighbouring hourly measurements by a set threshold value were removed. Subsequently, the data were aggregated to daily
values, with the exclusion of measurements that had been flagged for quality issues. For empirical estimation, equations 1 to
3 were used. Parameters beta, omega and psi were estimated by minimizing the residuals of the measurements at $t = 0$ (current
day) to $t = -1$ (previous day). Here, measurement at $t = -1$ is considered as reference for each time step $t = 0$ (or $t = 1$ as
denoted in Equations 1 to 3, respectively). The residuals were estimated by using the root mean squared error and by the mean
absolute error, the latter one because outliers receive lesser weight. Remaining uncertainty was assumed to be attributed to
changes in local hydrogen pools such as soil moisture, Poisson noise which is considerably small for large time intervals, and
measurement uncertainties of the environmental sensors. In the later presented synthetic case, Poisson noise and measurement
uncertainty of the environmental sensors can be excluded.

## 2.3   Uncertainty estimates

Uncertainty was quantified via moving block bootstrapping. Here, the observed time series were divided into 100 time
segments of equal length, with each block length representing one-seventh of the total time series.. The empirical scaling
parameters are estimated for each segment, parameter estimates were logged, and the uncertainty was defined as the standard
deviation of the parameter estimates calculated from these 100 bootstraps by sensor.

## 2.4   Synthetic test case

A synthetic test case was set up and used to test the optimization routine. The synthetic test case set up generates synthetic
neutron flux data that are used as 'truth' to test algorithm's performance under known conditions (Das et al., 2014; Pipunic et
al., 2008). For setting up a synthetic CRNS test case, incoming neutron intensity of the Jungfraujoch NMDB monitor, and soil





moisture, atmospheric pressure, and air humidity observations from the Merzenhausen test site, Germany (Bogena et al., 2018)
       were used to produce a synthetic neutron flux signal following the approach by Davies et al. (2022). In brief, time series of
       point scale soil moisture observations were used to generate synthetic neutron flux using a fixed $N_0$ = 1205 and the inverse
       relationship of neutron flux with soil moisture (Desilets et al., 2010):

$$SWC = a_0/\left(\left(N_{pih}/N_0\right) - a_1\right) - a_2 \qquad (4)$$


       Here, $N_{pih}$ is the synthetic corrected neutron flux, $a_0$, $a_1$, $a_2$ are empirical constants, and $N_0$ is a calibration parameter for
       reference conditions. This neutron flux is transformed to uncorrected neutron flux using the scaling equations (Eq. 1-3) and
       fixed reference conditions ($P_{ref}$ = mean atmospheric pressure, $abs_{ref}$ = 7 g/m3, $Inc_{ref}$ = 1.0). This results in a first CRNS time
       series of neutron flux that includes dynamics of soil moisture and environmental conditions. Poisson noise was added to the
hourly data to generate realistic noise for the second time series of neutron flux.

       The neutron observations were used in the synthetic scenario to estimate beta, omega and psi inversely using the previously
       described inversion routine. The proposed inverse estimation of beta, omega and psi neither is aware nor is made aware of
       changes in soil moisture. Thus, the soil moisture enters the calibration as an unknown and as a noise term. The inverse
parameter estimation results are reported. The synthetic scenario was run a) once without Poisson noise added, and b) 1,000
       times with individual hourly Poisson noise and soil moisture dynamics. The scenario b) resulted in 1000 results. The ensemble
       was used to calculate the percentage of parameter estimates outside the estimated parameter value +/-uncertainty.

### 2.5    Sensitivity analysis

       Sensitivity of soil moisture estimates from CRNS data are explored with numerical experiments. Sensitivity of soil moisture
estimates on scaling parameters is critical and one reason why improved scaling parameters are desirable. Here, we define
       three levels of true reference soil moisture for the numerical experiments: low (0.1 m³/m³), medium (0.25 m³/m³) and high (0.4
       m³/m³). For reference soil moisture, 'theoretically observed neutron flux' was calculated for a range of possible **empirical
       scaling parameters** as found for the COSMOS Europe sites in this study. This neutron flux is recalculated to estimated soil
       moisture assuming standard scaling parameters (beta = −0.0074, omega = −0.0054 and psi = 0.7). It should be noted that this
is not an accurate representation of the true reference soil moisture. This difference in soil moisture will be larger for **empirical
       scaling parameters** being more far from **standard scaling parameters**. It is important to note that estimates will not be error-
       free if the environmental conditions and/or scaling parameters do not align with the standard parameters. We then provide a
       heat map of the difference between the estimated soil moisture of standard parameters and the 'true' soil moisture, representing
       the error given different environmental conditions to illustrate the potential impact of differing site-specific scaling parameters
on soil moisture estimates.





## 2.6 Energy dependence of scaling parameters

In order to ascertain whether the energy spectrum of the CRNS detectors could be of significance, scaling parameters for the thermal neutrons that were measured using co-located bare detectors were also computed. The thermal neutrons are rather less
sensitive to hydrogen within the footprint and may show different scaling dependence on environmental factors (Jakobi et al., 2022). This results in potentially different scaling parameters for the thermal neutrons compared to epithermal neutrons used for soil moisture detection, although they are measured at the same location.

## 2.7 Model evaluation

The method developed in this study was evaluated in the Alento test-site which was chosen since the standard correction parameters were strongly different to those found in this study. The Alento River Catchment (ARC) is located in Campania, an administrative region situated in southern Italy. Recently, two experimental sub-catchments (MFC2 and GOR1) were instrumented with: *i*) a CRNS (CRS2000/B, Hydroinnova LLC, Albuquerque, USA; *ii*) a wireless sensor network (SoilNet, Forschungszentrum Jülich, Germany) controlling  a total of 40 GS3 sensors (METER Group Inc., Pullman, WA, USA)
monitoring soil water content at the soil depths of 15 cm and 30 cm over 20 positions around the CRNS; *iii*) a weather station to monitor rainfall, air temperature, relative humidity, wind speed, and net solar radiation (Nasta et al., 2024; Nasta et al., 2020). Three periods were selected out of the whole time series. These periods are featured with continuous measurements of neutron flux, atmospheric pressure, air humidity, incoming neutron intensity from Jungfraujoch and soil moisture by GS3 sensors. Selection criteria were measurement continuity, and for either period high variation of incoming neutron intensity, air
humidity, and atmospheric pressure, respectively. Soil moisture was vertically weighted using the approach proposed by Power et al. (2021). Calibration was performed for each time period individually using the site specific data reported in Bogena et al. (2022), mean vertically weighted soil moisture over the time period, and mean corrected neutron flux over this time period. For evaluation, the RMSE was calculated for CRNS soil moisture using the conservative parameters (reference approach) and using the new parameters presented in this paper. Both approaches are compared against weighted soil moisture.


## 3 Results

### 3.1 Synthetic test case

The results of the synthetic test case demonstrate that the parameters beta, omega, and psi can be accurately estimated. (Table 2). This also applies to the case of  dynamic soil moisture and additional Poisson noise. Parameter estimates without Poisson
noise are about as close to true values as estimates with Poisson noise (Table 2). No Poisson noise resulted in correct estimates of the true values for beta = −0.0074 +/− 0.00001, omega was estimated closely as 0.00535 +/−0.0001 and psi was also




estimated close to the synthetic truth as 0.716 +/−0.02. With Poisson noise, the differences between estimates and truth were slightly larger than for the case with soil moisture dynamics only. The uncertainty was notably higher with added Poisson noise compared to only soil moisture dynamics. Uncertainty values were calculated with moving block bootstrapping. The

synthetic test demonstrated that 96% of the parameter estimates were within the uncertainty range to the synthetic truth. Moreover, the mean values of the 1,000 realizations were within the uncertainty ranges. Out of the 1,000 realizations, the percentage of estimates outside the uncertainty range was about equal for all three parameter sets. These uncertainty bounds will be reported in the further analysis.

**Table 2: Parameter estimation results for synthetic experiments with 1,000 realizations. The results for 'SWC+Poisson noise' are mean parameter estimates of the 1,000 realizations. The percentage of parameter estimates inside the uncertainty bounds with respect to 1000 realizations is reported.**

|  | Beta estimated | Beta uncertainty | Omega estimated | Omega uncertainty | Psi estimated | Psi uncertainty |
|---|---|---|---|---|---|---|
| Synthetic truth: | −0.0074 |  | 0.0054 |  | 0.7 |  |
| SWC dynamics: | −0.00741 | 0.00001 | 0.00534 | 0.0001 | 0.715 | 0.02 |
| SWC+Poisson noise: | −0.00741 | 0.00023 | 0.00532 | 0.0013 | 0.711 | 0.15 |
| Within uncertainty: | 96 % |  | 96 % |  | 96 % |  |

## 3.2    Beta estimates (atmospheric pressure scaling)

Beta estimates for the sites of the COSMOS Europe dataset excluding site LEC001 ranges between −0.0052 and −0.0078 with mean and median of −0.0071 and −0.0073, respectively (Figure 1). These are parameter estimates for sites at cutoff rigidities smaller than root mean square error as convergence criteria and using correction for incoming neutron intensity from Jungfraujoch. Beta estimates were also estimated using incoming neutron intensity correction with Oulo, Apty, Mexico and Athens NMDB monitors. Moreover, the beta parameter estimates and uncertainties for these NMDB monitors (Table 1) were

very close to those obtained with Jungfraujoch data. In general, the Pearson correlation coefficient was high (larger 0.9) which indicates that beta estimates are rather indifferent to the choice of NMBD monitor. Error bars indicate the uncertainty of beta estimates which is 0.00036 on average for the sites (Figure 1). However, for sensors with cutoff rigidities larger than 4.5 we obtained beta estimates between −0.006 and −0.007 even with uncertainties considered. This confirms a close relationship of beta and cutoff rigidity although the beta values are far from the previously estimated range between −0.007 and −0.008. The

relationship of beta estimated with this method and beta estimated by the method of McJannet and Desilets (2024) also shows an $R^2$ of 0.46. It is notable that the range of beta estimates in this study is considerably broader than that observed in previously published beta estimates or the commonly utilized reference value of -0.0076.



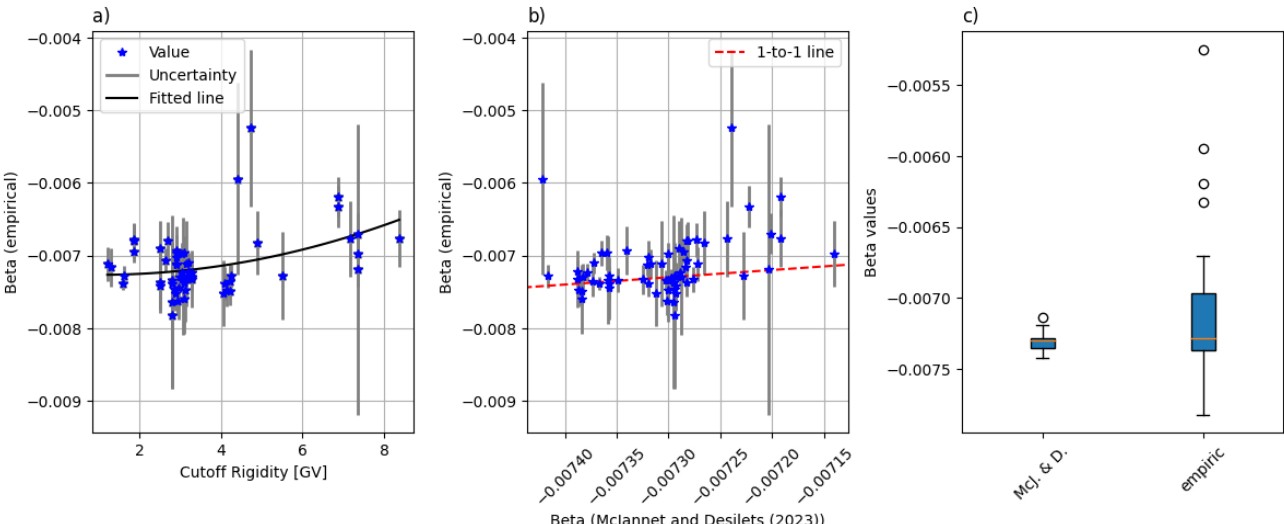

**Figure 1: a) Beta estimate and 2$^{nd}$ order polynomial regression with cutoff rigidity for the COSMOS Europe sites, excluding sites BUC001 and LEC001; b) Beta estimates of this study (empirical) in comparison to those derived by McJannet and Desilets (2023); c) Boxplot of beta estimates of this study (empiric) in comparison to those derived by McJannet and Desilets (2023). Indicated are the median of the data sets (horizontal bar), the outliers (circles), the box (25$^{th}$ and 75$^{th}$ percentile – interquartile range), and whiskers (1.5 x interquartile range).**

**3.3     Psi estimates (incoming neutron intensity scaling)**

Psi estimates showed a strong dependence on cutoff rigidity if, for example, Jungfraujoch station (CR=4.5 Giga Volt, GV) was used for incoming neutron intensity correction (Figure 2a). Here, psi ranged between 0.05 and 1.12. CRNS sites with cutoff rigidity close to Jungfraujoch exhibited higher psi than those with cutoff rigidity different to Jungfrauhoch (CR<2 and CR>6). Although not all sites with CR between 2.5 and 4.5 had psi equal to one, a site located in the Alps in vicinity to

Jungfraujoch exhibits psi equal 1. This indicates a 1:1 linear scaling of incoming neutron intensity with neutron intensity measured at the CRNS site. The estimates of psi also indicate that cutoff rigidity is a significant factor in defining psi. It should be noted, however, that the elevation of the NMDB monitor and geographical distance may also have an impact  in defining psi.




**Figure 2: Psi estimated for 64 COSMOS Europe sites and different NMDB monitors: (a) Jungfraujoch, (b) Apty and Oulu. Figure c) includes also results for Lmks, Mexico and Athens with NMDB monitor's cutoff rigidity denoted in the legend. Psi estimates are provided as markers dot, x and asterisk. Polynomial regressions to NMDB monitors are shown as lines. Hawdon et al. (2014) is provided with reference to Jungfraujoch.**


Mean psi estimates for all sites using either of the sites Jungfraujoch (CR=4.5 Giga Volt, GV), Oulu (CR=0.8 GV), Apty (Russia, CR=0.65 GV), Mexico (CR=8.3 GV) or Athens (CR=8.53 GV) were 0.62, 0.74, 0.74, 0.95 and 0.86, respectively – indicating a strong influence of incoming neutron intensity on CRNS signal (Figure 2c). However, correlation of psi values for different stations was not always strong. For example, Jungfraujoch exhibited the highest correlation (r = 0.45) with the

APTY monitor. Overall, highest correlation was observed between Apty and Oulu monitors (r=0.87). These rather low





correlations indicate differences with regard to psi estimated for individual NMDB monitors. Correlation between Lmks and Jungfraujoch was particularly low (r = 0.13) despite both monitors are located at high altitude (+2000 meter above sea level) and in Central Europe. Moreover, psi estimates for Athens and Mexico monitors correlated only weakly with 0.28 despite a low difference (0.2 GV) in cutoff rigidity of Athens and Mexico's. Common to all CRNS sites is that psi is smaller for sites

CRNS sites with large cutoff rigidity. However, the psi differs depending on NMDB monitor and location.

### 3.4    Omega estimates (air humidity scaling)

Estimates of omega range between −0.016 and 0.017, with mean and median at -0.0061 and −0.0066, respectively (Figure 3). Here, as well as for the estimation of the other environmental factors, data quality plays a crucial role. Omega showed a

remarkable large range. Omega's mean (−0.0065) and median (−0.0068) are close but not equal the originally estimated value omega (−0.0054) from calculations with a Monte Carlo Neutron Particle model (Rosolem et al., 2013). Standard deviation shows a large uncertainty of 0.0041 for omega. Standard deviation diminishes to 0.0018 if the three highest and three lowest estimates of omega are removed from the dataset with 64 sites.

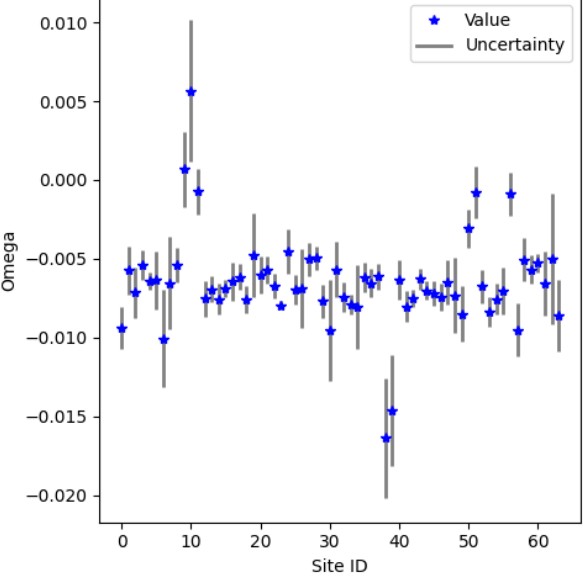

**Figure 3: Omega estimates (blue star) and uncertainty (grey bars) of omega estimates for scaling CRNS counts with air humidity. Result of the site WEC001 is excluded because of its high uncertainty (uncertainty = 0.02).**



## 3.5    Sensitivity of soil moisture to scaling parameters

The results for beta, psi and omega showed significant differences amongst sites and to reference values. The sensitivity

analysis of soil moisture depending on scaling parameters demonstrates that difference between true and estimated soil moisture can be easily four volumetric percent. The error, i.e., different in estimated soil moisture to true soil moisture, depends on three factors: reference soil moisture, scaling parameter, and change in environmental variable (atmospheric pressure, air humidity, incoming neutron intensity; see Figure 4). Reference values result in error-free estimates i.e. difference of estimated soil moisture while any change of the factors results in difference between truth and estimated soil moisture. Reference values

and the two percent difference are highlighted by the black line in Figure 4. The sensitivity analysis demonstrates that differences matter more for high soil water content. Scaling factors and reference values strongly matter for soil moisture estimates. Generally, less differences can be expected for scaling factors chosen at medium level and average environmental calibration conditions for atmospheric pressure, air humidity and incoming neutron intensity. The heat maps (Figure 4) indicate strongest differences if scaling parameters and calibration conditions are at the far end of either side.






**Figure 4: Sensitivity of soil moisture calculated as difference between estimated and true soil moisture. Estimated soil moisture depends on atmospheric pressure (a,b,c), incoming neutron intensity (d, e, f) and air humidity (g, e, h) and there respective ranges. True soil moisture remained constant while estimated soil moisture depends also on reference value of omega and reference value of air humidity. Contour lines show soil moisture differences of 0.00 and 0.02 m³/m³ to reference values. Differences of estimated to true soil moisture were always highest for moist conditions e.g. 0.4 m³/m³.**

## 3.6    Uncertainty for data driven parameter estimates

We found that the parameter estimates strongly depend on data quality and data availability. The following Figure 5 shows the uncertainty of beta with regard to days observed (a) and with regard to total neutron counts (b). Both metrics show a strong correlation with threshold values that can be identified to generally constrain uncertainty of the beta estimate. The same holds for omega and psi estimates. Given the uncertainty depending on days of measurement and overall observed neutron counts,





1,000 consecutive days of observation or 20,000,000 neutron counts appear to result in rather low uncertainty of scaling parameters. Here, the slope of the polynomial approximation flattens, indicating a plateau that is reached from these values.

The results for the synthetic experiment were always better than the regression based on observed data.

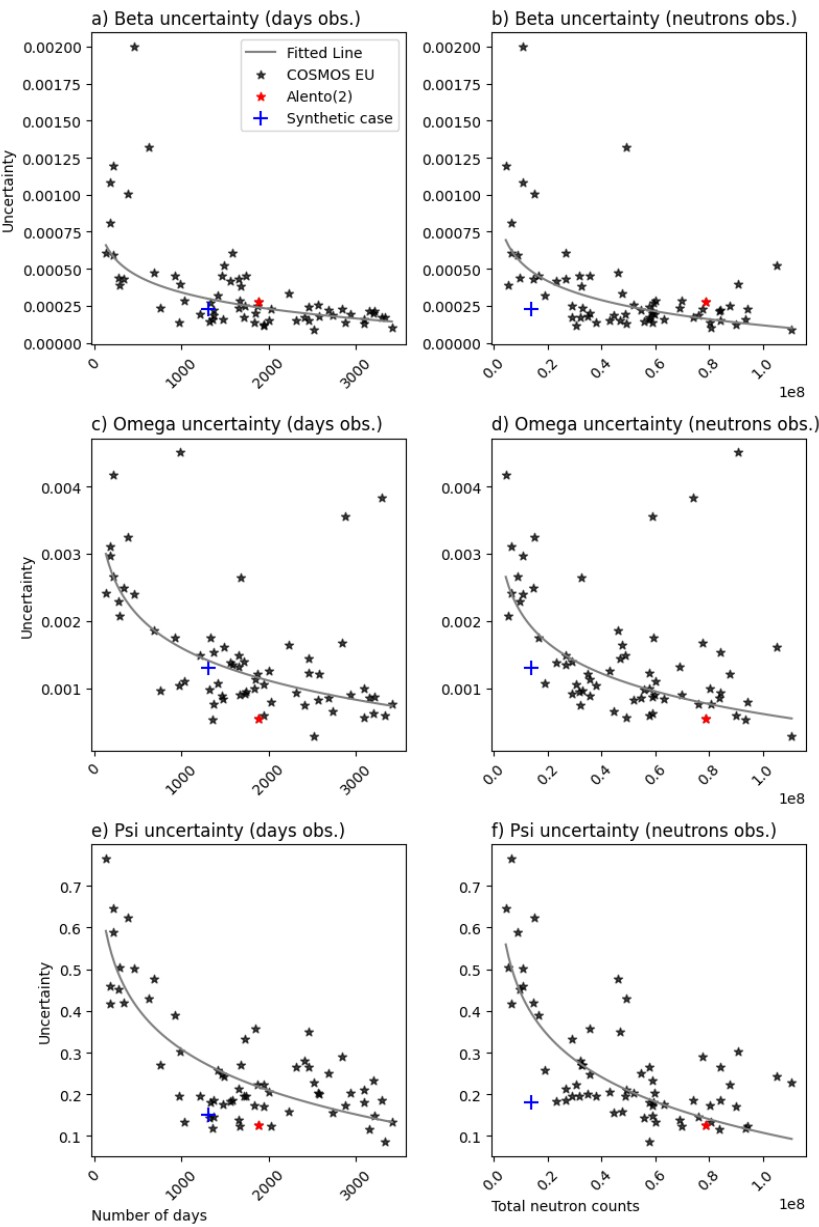

**Figure 5: Uncertainty of beta, psi, and omega with respect to consecutive days of measurement (a, c, e) and neutron flux (b, d, f) over the whole time period. One outlier was removed for omega. Uncertainty calculated for the Alento site is marked as red asterisk, the synthetic test case (1000 realizations) is marked as blue plus. The solid gray lines denotes the logarithmic fitted line.**






## 3.7    Energy dependence of scaling parameters

Twelve CRNS sensors provided data of a different energy spectrum than neutrons used for soil moisture. Accuracy and number of detectors did not allow to establish a clear relationship between cutoff rigidity and scaling parameter. Figure 6 shows the

results of scaling parameters for epithermal and thermal neutron data. The results demonstrate for beta a mean absolute difference of 0.0004 mostly subject to one of twelve sensors. Omega for thermal neutron counts is smaller than for moderated neutron counts with a mean absolute error of 0.0036. This is a clear indication that thermal neutron counts are less sensitive to air humidity changes than moderated neutrons. The scaling factor for incoming neutron intensity psi is also smaller for thermal neutron counts than for epithermal neutron counts which indicates a smaller impact of the incoming neutron intensity on

thermal neutrons.

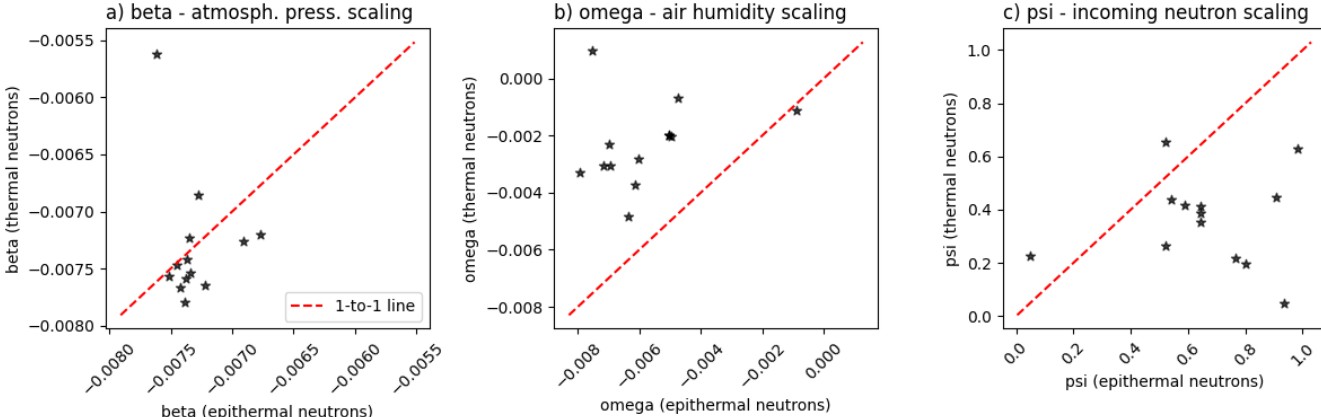

**Figure 6: Results of scaling parameters, namely a) beta, b) omega, and c) psi, for thermal neutron counts using bare counter tubes against epithermal neutron counts using a moderated tube. Moderated i.e. epithermal counts are commonly used for soil moisture determination with CRNS.**


## 3.8    Model evaluation

The evaluation results are reported in Figure 7 for the MFC2 experimental field (named ALC002 site in Bogena et al., 2022) in the Alento site. In all three cases, the new approach showed slightly lower RMSE values compared to the reference standard approach. Although the error is rather small for all methods, the results provide insights on the reasonability of the parameter

values obtained and potential to outperform the reference approach. For the three periods, the new approach improves the RMSE by 28 %, 25 % and 25 %, respectively (Figure 7).



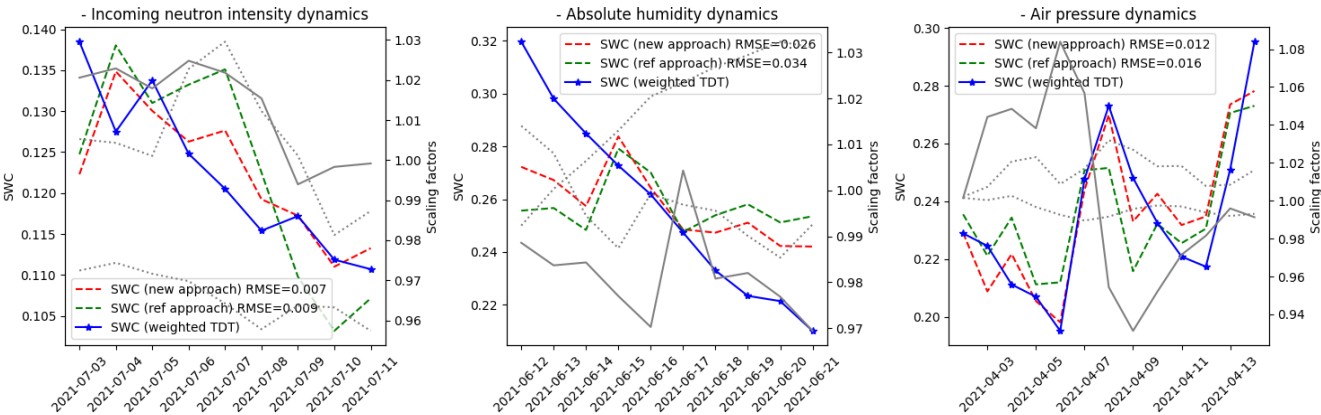

**Figure 7: Evaluation results for MFC2 in the Alento site (ALC002 in Bogena et al., 2022). High variation of a) incoming neutron intensity, b) absolute air humidity and c) atmospheric pressure are compared against observed vertically weighted soil water content (SWC). Grey solid line are the respective correction factors, with the other dashed lines representing the corresponding secondary scaling factors for this period.**

## 4    Discussion

### 4.1    Overview and interpretation of key findings

This study evaluated the estimation of scaling parameters – beta, psi, and omega – used in cosmic ray neutron probes (CRNS) for measuring soil moisture, with an emphasis on the strengths and limitations of a data-driven approach compared to traditional analytical methods. The main motivation behind this research was to refine soil moisture estimation by integrating both data-driven and analytical techniques, recognizing the necessity of a hybrid approach that balances complexity, accuracy, and uncertainty.

The results indicate that these scaling parameters can be well estimated using observational data alone, without the need for direct soil moisture information, providing a robust alternative to purely analytical methods. The proposed methodology offers a promising new tool for refining scaling parameters, potentially improving the ability to differentiate between site-specific characteristics. Thus, this data-driven approach may serve as either an alternative to or complement analytical methods developed in previous studies, supporting a hybrid approach that incorporates both data-driven and analytical scaling functions. More detailed interpretations of the results will be discussed in the following subsections and summarized in the conclusion.





## 4.2 Scaling parameter estimation results

### 4.2.1 Atmospheric pressure and neutron flux

The synthetic test case demonstrated that the parameters beta, omega, and psi can be reliably estimated, with uncertainties quantified to provide 96% accuracy in parameter estimates. In real-world conditions, results from the COSMOS Europe dataset confirmed the relationship between beta and cutoff rigidity. A multiple linear regression analysis revealed that mean atmospheric pressure, site altitude, and cutoff rigidity together explained 52% ($R^2$) of the variability in beta estimates. This finding aligns with previously published research (Desilets et al., 2010; Clem and Dorman, 2000; Dorman, 2004), which

reinforces confidence in the method's validity.

In contrast to many earlier studies, this research derived beta values for the energy spectrum of CRNS sensors, which detect particles with a different energy spectrum compared to traditional neutron monitors. Few studies have successfully analysed beta using direct data from CRNS sensors (Schrön et al., 2024), with most focusing on scaling parameters derived from neutron

monitors (Clem and Dorman, 2000; McJannet and Desilets, 2023; Desilets et al., 2006) and analytical models (Zreda et al., 2008; Köhli et al., 2023; Desilets et al., 2010). In comparison to these previous studies, this research found a wider range of beta values, particularly at high cutoff rigidities and high altitudes.

This larger range of beta estimates has important implications for soil moisture estimation from neutron flux, as it indicates a

significant sensitivity of beta to environmental factors. The broader range of beta values observed, even for thermal neutrons, underscores the influence of the energy spectrum of the observed neutrons (Bütikofer, 2018). These findings suggest that the energy spectrum plays a critical role in determining beta values, which differ significantly from those derived using neutron monitors.

Future research on atmospheric pressure scaling should aim to further investigate site-specific and sensor-specific characteristics to improve the development of analytical scaling functions. Identifying these factors could enhance the precision of soil moisture estimates across different environments and sensor types.

### 4.2.2 Air humidity and neutron flux

The air humidity scaling parameter, omega, closely aligned with values proposed in other studies (Köhli et al., 2021; Rosolem et al., 2013). However, the mean omega value found in this study (−0.0065) differed by approximately 20% from the value proposed by Rosolem et al. (2013), which could have a significant impact on soil moisture estimates under varying air humidity conditions compared to reference environments.



Our analysis using thermal neutron detectors confirmed the validity of omega estimates, with omega values being smaller, consistent with the energy spectrum of these sensors. This is due to their reduced sensitivity to hydrogen within the vertical air column and the sensor's footprint area. Similar findings were reported by Schrön et al. (2024) and (Rasche et al., 2023), indicating that thermal neutrons exhibit a diminished sensitivity to air humidity.

In contrast to some previous studies, our results align more closely with those of Köhli et al. (2021), who identified a stronger influence of air humidity on neutron flux scaling in CRNS. The omega values reported here were higher than those found in other studies who identified a stronger impact of air humidity on scaling neutron flux of CRNS. Here, the values were higher than those of other studies (Schrön et al., 2024; Rosolem et al., 2013) , resulting in a steeper slope (−0.0065) and potentially different functional form for the air humidity impact on neutron flux scaling.


### 4.2.3    Incoming neutron intensity and neutron flux

The results demonstrate that cutoff rigidity significantly influences the estimation of scaling parameters, particularly psi, which scales incoming neutron intensity. For example, psi values tend to be higher at sites where the cutoff rigidity is similar to that of reference NMDB stations, such as Jungfraujoch (4.5 GV). Sites with cutoff rigidities outside this range, however, exhibit
more variability in psi estimates. This suggests that proximity in cutoff rigidity between a CRNS site and its corresponding NMDB reference station is critical for accurate scaling of neutron intensity measurements.

Nevertheless, other factors, including elevation and geographic distance, also affect the scaling impact. For instance, although Lmks and Jungfraujoch are located at similar altitudes, they show different scaling impacts, highlighting that cutoff rigidity is
not the only determinant. This finding implies that while cutoff rigidity is a key factor, a more comprehensive approach that accounts for elevation, geographic proximity, and local environmental conditions at both CRNS and NMDB sites is required for accurate analytical parameter estimation (Bütikofer, 2018; Gerontidou et al., 2021; McJannet and Desilets, 2023).

A data-driven approach may offer a viable alternative to these analytical models, potentially allowing for the selection of
NMDB monitors that are more appropriate for some CRNS sites than the commonly used Jungfraujoch station. This flexibility could improve scaling accuracy by incorporating NMDB monitors that are geographically closer or better suited to the environmental conditions of the CRNS site (Bogena et al., 2022; Zreda et al., 2012).

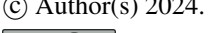



## 5 Conclusions

The results of this study demonstrate that the new calibration method for estimating scaling parameters (beta, psi, and omega) in cosmic ray neutron probes (CRNS) is both reliable and robust. However, it should be noted that the performance of this method is dependent on the quality of the data used. The reliability of the method is supported by the strong correlations between the estimated parameters and those predicted by analytical approaches. However, the study also indicated that there are larger uncertainties than previously assumed, and that calibration parameters may differ significantly from the standard

values. The observed relationships between parameter values, cutoff rigidity, and elevation provide further validation the approach. However, the uncertainties highlight the need for careful data selection and parameter estimation to ensure the reliability of the results.

Furthermore, the results indicate that pressure- and efficiency-corrected data from NMDB sensors located near the CRNS

sensor, particularly those with similar or lower cutoff rigidity, should be given preference. This result is in line with existing analytical methods and demonstrates that cutoff rigidity is not a sufficient condition for optimal scaling parameterization. The study further underscores the importance of site- and sensor-specific scaling parameters to guarantee precise soil moisture estimates. Sensor-specific attributes, such as the energy spectrum monitored, significantly influence the accuracy of these estimates. While cutoff rigidity and elevation were identified as critical factors influencing beta, psi, and omega, additional,

unidentified factors may also play a role. To ensure accuracy, it is recommended that periods with average environmental conditions be selected for calibration to minimize discrepancies between estimated and actual soil moisture.

## 6 Outlook and future directions

Future research should focus on enhancing data collection methods, defining quality standards, quantifying parameter uncertainty, and increasing the length of observation periods to reduce uncertainty. Additionally, refining scaling methods to

better account for the energy dependence of neutrons, geographic and environmental factors, and other site- and sensor-specific conditions that influence scaling parameters is necessary. Although this study has identified key principles for scaling, further fine-tuning is required to fully understand and quantify the scaling functions. The findings revealed a higher variability in beta, a greater impact of air humidity on CRNS neutron intensity, and more variation in scaling factors for NMDB monitors than previously expected.


The proposed method can be tested across a broader range of sites and conditions to explore its full potential and limitations. Moving forward, a hybrid calibration approach combining the benefits of both analytical and data-driven methods may offer the most feasible solution. Such an approach would balance known theoretical relationships with sensor- and site-specific characteristics, while also minimizing the calibration period required. By carefully considering site-specific conditions,

environmental factors, and data quality, the data-driven method can improve the accuracy and reliability of soil moisture



estimates using CRNS. Future research should continue to refine these calibration techniques and further explore the factors that affect scaling functions for accurate soil moisture estimates at field scale.

## 7 Competing interests

The authors declare that they have no conflict of interest.

## 8 Acknowledgements

We acknowledge the NMDB database (www.nmdb.eu), founded under the European Union's FP7 programme (contract no. 213007) for providing data. We thank the Federal Ministry of Education and Research (BMBF) for financing of the WASCAL CONCERT project under WRAP 2.0, grant number "01LG2101A". The monitoring activities in the Alento Hydrological Observatory were supported by the MiUR-PRIN-PNRR-2022 Project "Assessing and mapping novel agroecosystem
vulnerability and resilience indicators in southern Italy-ASAP" (grant P20228L3KX).

## 9 Code Availability

The code for this method and data processing is available via github.com/zalf-rpm/CRNS-Scaling.

## 10 Data availability

Data for incoming neutron monitors is available at the website www.nmdb.eu. Data of the COSMOS Europe CRNS network
is available via https://doi.org/10.5194/essd-14-1-2022 (Bogena et al. 2022) and at

https://teodoor.icg.kfa-juelich.de/ibg3butt/ibg.butt.download?FileIdentifier=519e0691-7eb3-4351-aff5-c0a0335933ab

## 11 Author contribution

Formal analysis and writing – original draft preparation: RB. Conceptualization, writing – review and editing: RB, PD, PN, and HB. All authors have read and agreed to the submitted version of the paper.




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
