# Peer review of "Data-Driven Scaling Methods for Soil Moisture Cosmic Ray Neutron Sensors"

_EGUsphere, 2024_

## Author Comment (AC1)

The reviewer's comment is written in bold. The reply of the authors is written in non-bold.

**The Baatz et al. (2024) paper presents a very clear explanation of all the scaling factors used to correct cosmic ray neutron counts. In particular, sections 2.1.1 to 2.1.3 provide great detail on each. The intro paragraph at line 54 presents each very succinctly. I appreciate their efforts to really illustrate these concepts - and the fact that many of us have inherently considered these essentially fixed parameters.**

**The authors use inverse modeling to derive model parameters (e.g., beta, omega and psi) and their uncertainty. However, it is a little unclear what the forward model is they are inverting. Equations 1-3 and multiplied to get the total flux correction (Npih, eq. 4) at Line 186. The synthetic experiments are presented well. I am not following inversions of beta, omega and psi at the site level. Could you present the forward model and the error term that is being minimized? Or, if I am off target with the optimization scheme, could you elaborate on the inversion routine a little more?**

Thank you for your thoughtful feedback and for highlighting the need to better clarify the inversion process. We will address this by modifying Section 2.2 "Inversion of Scaling Functions." Specifically, we propose removing lines 163–167 and replacing them with the following explanation of the forward model and inversion routine:

"The forward model used for estimating the parameters beta, omega, and psi is based on the combination of scaling functions for atmospheric pressure, absolute air humidity, and incoming neutron intensity, as detailed in Equations (1), (2), and (3). The forward model computes the neutron flux N at time t by applying these scaling factors to the observed neutron flux $N_{t-1,obs}$ of the previous time step (t-1). This previous time step essentially serves as reference condition:

$$N_{t,est} = N_{t-1,obs} \times exp\{beta(P_t - P_{t-1})\} \times \{1 + omega \times (abs_t - abs_{t-1})\} \times \{1 + psi \times (Inc_t - Inc_{t-1})\} \qquad (4)$$

Parameters beta, omega and psi are the free parameters to be optimized. N, P, abs and Inc represent vectors of n days, and $N_{t,est}$ is the neutron flux estimated by using the corrections.

To optimize the three parameters, we use an inversion approach that minimizes the root mean square error (RMSE) between the observed neutron flux $N_{t,obs}$ and the estimated neutron flux $N_{t,est}$:

$$RMSE = \sqrt{\frac{1}{n} \sum_{t=1}^{n} (N_{t,obs} - N_{t,est})^2} \qquad (5)$$

where n represents the total number of days."

We will continue Section 2.2 as in the manuscript from line 167 onwards with:
"Remaining uncertainty was assumed […]."

---

## Author Comment (AC2)

The reviewer's comment is written in bold. The reply of the authors is written in non-bold.

**This manuscript presents a novel data-driven approach for calibrating scaling parameters used in cosmic-ray neutron sensing (CRNS) for soil moisture measurements. The work makes a meaningful contribution to improving CRNS methodology, though there are some aspects that would benefit from revision. For example, more detailed discussion of the practical implementation of the new approach can be made, what the implications of the new methods for CRNS users can be discussed.**

Thank you for the positive overall evaluation of the manuscript and for agreeing with us on the novelty of the calibration approach. We appreciate your time and effort in reviewing the manuscript. In the forthcoming revision we will consider each of your suggestions and implement the necessary changes.

With regard to the comment on practical implementation of the new approach, we suggest the addition of a new discussion subsection that seamlessly integrates the last paragraph of the discussion:

"4.3     Implementation and implications of the new approach
A data-driven approach may offer a viable alternative to semi-analytical models and offers practical benefits for CRNS users. For instance, it allows for the selection of NMDB monitors that are more suitable for specific CRNS sites than the commonly used Jungfraujoch station. This flexibility allows for improved scaling accuracy by incorporating NMDB monitors that are geographically closer or better suited to the environmental conditions of a given CRNS site (Bogena et al., 2022; Zreda et al., 2012). Additionally, our method can be seamlessly integrated into existing CRNS workflows, complementing traditional methods to improve calibration reliability. By improving calibration accuracy, our approach supports robust soil moisture estimates, enabling better-informed decisions in agriculture, hydrology, and climate monitoring."

**The manuscript is well-structured and generally well-written, though some sections could be more concise and clear. The abstract could better highlight the quantitative improvements achieved over existing methods, the current version of abstract lacks detailed descriptions using some values. In the methods section, I think it would benefit from a general paragraph summarizing what it takes to complete the calibration using the new method, maybe a flowchart can be added.**

Thank you for highlighting the need of a summarizing paragraph of what is required to complete the calibration. We therefore add a brief paragraph at the beginning of Section 2, in order to allow the reader more easily to access the new scaling approach:

"This study introduces a data-driven method for estimating the scaling parameters beta, psi, and omega in cosmic ray neutron sensing (CRNS) to improve soil moisture measurement accuracy. Section 2.1 outlines the scaling parameters, which correct for atmospheric pressure, incoming neutron intensity, and absolute air humidity. The forward model, detailed in Section 2.2, combines these scaling functions to estimate neutron flux by applying the corrections to the observed flux from the previous time step. Uncertainty estimates, described in Section 2.3, are calculated using bootstrapping techniques to evaluate the robustness and reliability of scaling functions. Together, this integrated approach provides a systematic and flexible framework for site- and sensor-specific calibration."

We further present a more complete description of the objective function for the inversion routine which is also in line with the comment by Todd Caldwell:

"The forward model used for estimating the parameters beta, omega, and psi is based on the combination of scaling functions for atmospheric pressure, absolute air humidity, and incoming neutron intensity, as detailed in Equations (1), (2), and (3). The forward model computes the neutron flux N at time t by applying these scaling factors to the observed neutron flux $N_{t-1,obs}$ of the previous time step (t-1). This previous time step essentially serves as reference condition:

$$N_{t,est} = N_{t-1,obs} \times exp\{beta(P_t - P_{t-1})\} \times \{1 + omega \times (abs_t - abs_{t-1})\} \times \{1 + psi \times (Inc_t - Inc_{t-1})\} \tag{4}$$

Parameters beta, omega and psi are the free parameters to be optimized. N, P, abs and Inc represent vectors of n days, and $N_{t,est}$ is the neutron flux estimated by using the corrections.

To optimize the three parameters, we use an inversion approach that minimizes the root mean square error (RMSE) between the observed neutron flux $N_{t,obs}$ and the estimated neutron flux $N_{t,est}$:

$$RMSE = \sqrt{\frac{1}{n}\sum_{t=1}^{n}\left(N_{t,obs} - N_{t,est}\right)^2} \tag{5}$$

where n represents the total number of days."

**In all, this paper represents a valuable contribution to the field and is suitable for publication in HESS after moderate revision.**

Thank you. We will implement the suggested changes accordingly.

**Specific line-by-line comments:**

**Lines 87-93: The objectives should be stated more explicitly here. Consider put specific research questions and objectives at the beginning.**

We thank the reviewer for this constructive suggestion. We will rephrase the last paragraph to:

"This study aims to address these limitations by presenting a novel data-driven, empirical approach for calibrating scaling parameters (i.e., beta, psi, and omega) used in CRNS. Specifically, this study has three objectives: (1) to develop an inverse method that directly calculates correction parameters from measurement signals while treating soil moisture dynamics as a noise term, (2) to evaluate the accuracy of current scaling functions, and (3) to quantify the impact of local environmental factors on calibration parameters. The hypothesis is that this approach, by accounting for site-specific and sensor-specific conditions, will improve the accuracy and reliability of CRNS soil moisture measurements. By improving the accuracy of soil moisture determination, this study contributes to better informed decisions in agriculture, hydrology, and climate monitoring."

**Lines 315-324: The sensitivity analysis results could be more quantitative when describing impact of parameter variations on soil moisture estimates. Consider refer to specific values (sometimes can be in brackets after your statements) in this section.**

We agree and will be more quantitative in the description. We will add specific values accordingly.

**Grammar Issues:**

**Line 74: "effect" to "affect"**

Thank you. We changes this.

**Line 142: Missing space after absref**

Thank you. We changes this.

**Line 330: "Contour lines show soil moisture differences of 0.00 and 0.02 m³/m³ to reference values." Consider rephrasing to avoid ambiguity**

Thank you. We clarified:
"Contour lines show soil moisture differences of 0.02 m³/m³ (curved) and 0.00 m³/m³ (straight) to reference values."

---

## Author Comment (AC3)

The reviewer's comment is written in bold. The reply of the authors is written in non-bold.

**The manuscript presents a novel approach to calibrate CRNS for soil moisture estimation using a data-driven, inverse modeling technique. This method promises to address site-specific and sensor-specific variations that are often overlooked in traditional analytical methods. The idea is innovative and highly relevant to the field.**

Thank you for the positive overall evaluation of the manuscript and for agreeing with us on the novelty of the calibration approach. We appreciate your time and effort in reviewing the manuscript. In the forthcoming revision we will consider each of your suggestions and implement the necessary changes.

**However, there are several areas where the manuscript could be improved:**

**Comment:**

**1. The manuscript briefly talked about traditional analytical methods, but including additional details about these in methodology section would enhance its comprehensiveness. A short summary or explanation of how traditional methods are practically implemented, alongside their limitations, could enrich the manuscript significantly. Similarly, further clarification on the forward model used in the inverse method (possibly included in response to the first reviewer's comment) would be beneficial. This inclusion should highlight how analytical method calculate the soil moisture and how the newer approach will calculate. These short or one paragraph would be especially helpful for new readers to better understand the methodology.**

We agree with the reviewer that the points stated will improve the accessibility of the novelty to readers.

Starting with the forward model, we present a more complete description of the objective function for the inversion routine which is also in line with the comment by Todd Caldwell:

"The forward model used for estimating the parameters beta, omega, and psi is based on the combination of scaling functions for atmospheric pressure, absolute air humidity, and incoming neutron intensity, as detailed in Equations (1), (2), and (3). The forward model computes the neutron flux N at time t by applying these scaling factors to the observed neutron flux $N_{t-1,obs}$ of the previous time step (t-1). This previous time step essentially serves as reference condition:

$$N_{t,est} = N_{t-1,obs} \times exp\{beta(P_t - P_{t-1})\} \times \{1 + omega \times (abs_t - abs_{t-1})\} \times \{1 + psi \times (Inc_t - Inc_{t-1})\} \tag{4}$$

Parameters beta, omega and psi are the free parameters to be optimized. N, P, abs and Inc represent vectors of n days, and $N_{t,est}$ is the neutron flux estimated by using the corrections.

To optimize the three parameters, we use an inversion approach that minimizes the root mean square error (RMSE) between the observed neutron flux $N_{t,obs}$ and the estimated neutron flux $N_{t,est}$:

$$RMSE = \sqrt{\frac{1}{n} \sum_{t=1}^{n} (N_{t,obs} - N_{t,est})^2} \tag{5}$$

where n represents the total number of days."

We further will add a more detailed explanation on how traditional methods are implemented. This will briefly touch on the limitations of the traditional methods as well. Briefly, because it is elaborated in more detail in the introduction. However, it is needed to note these limitations in the methods section as well. We will add:

"2.1.1   Scaling parameters

Traditional semi-analytical methods estimate scaling parameters for air humidity, atmospheric pressure, and incoming neutron intensity primarily using Monte Carlo neutron particle simulations, limited CRNS measurement data, and NMDB data (see e.g. Köhli et al., 2023; Desilets et al., 2010; Dorman, 2004; McJannet and Desilets, 2023; Rosolem et al., 2013; Desilets and Zreda, 2003). These approaches laid the foundation for soil moisture estimation from CRNS by providing generalized scaling parameter estimates. However, they rely on strong correlations with global variables such as cutoff rigidity, latitude, and elevation, using data from relatively few reference stations scattered across the globe. While effective for global first estimates, these methods are limited in their ability to account for site-specific and sensor-specific characteristics, potentially resulting in inaccuracies in soil moisture estimation. In contrast, we propose a data-driven approach that directly calculates scaling parameters from observational data, enabling robust calibration tailored to local conditions, as detailed in the following subsections."

**2. The abstract could be improved by adding the exact results. Specifically mentioning the strong correlations and higher variability.**

Thank you for noting the need to clarify the abstract and include detailed results. We will change it towards:

"Cosmic ray neutron sensors (CRNS) are state-of-the-art tools for field-scale soil moisture measurements, yet uncertainties persist due to traditional methods for estimating scaling parameters that often lack the appropriate ability to account for site-specific and sensor-specific characteristics. This study introduces a novel, data-driven approach to estimate key scaling parameters (beta, psi, and omega) by directly calculating them from measurement data, emphasizing local environmental factors and sensor attributes. The method demonstrates reliability and robustness, with strong correlations between estimated scaling parameters and environmental factors such as cutoff rigidity, latitude, and elevation, and consistency with semi-analytical methods, such as an $R^2$ of 0.46 for beta. The study also reveals systematically higher variability in calibration parameters than previously assumed, highlighting the importance of this method, data quality, and careful selection of NMDB reference sites. The new method reduced RMSE by up to 25%, with differences in soil moisture estimates between traditional and data-driven methods easily reaching 0.04 m³/m³ and up to 0.12 m³/m³ under certain conditions. The sensitivity analysis showed that soil moisture estimation is most influenced by scaling parameters in the wet end of the soil moisture spectrum. We anticipate that the improved soil moisture accuracy achieved with our method will lead to better decisions in agriculture, hydrology, and climate monitoring. Future research should focus on further refining these scaling methods to increase the quality of CRNS data in order to further improve the accuracy of CRNS-based soil moisture estimates."

**Overall, the research article is well-written and presents a novel and valuable contribution to CRNS calibration methods.**

Thank you.